

# Effects of Intermittent Aerosol Forcing on the Stratocumulus-to-Cumulus Transition

Prasanth Prabhakaran[1,2], Fabian Hoffmann[3], and Graham Feingold[2]

[1]Cooperative Institute for Research In Environmental Sciences (CIRES), University of Colorado, Boulder, CO, USA
[2]Chemical Sciences Laboratory, National Oceanic and Atmospheric Administration, Boulder, CO, USA
[3]Ludwig-Maximilans-Universität München, Meteorologisches Institut, Munich, Germany

**Correspondence:** Prasanth Prabhakaran (prasanth.prabhakaran@noaa.gov)

**Abstract.** We explore the role of intermittent aerosol forcing (e.g., ship tracks, or injections associated with marine cloud brightening) on the stratocumulus-to-cumulus transition (SCT). We simulate a three-day Lagrangian trajectory in the north-east Pacific using a large-eddy simulation model coupled to a bin-emulating, two-moment, bulk microphysics scheme that captures the evolution of aerosol and cloud droplet concentrations. By varying the background aerosol concentration, we consider two
baseline systems - pristine and polluted. We perturb the baseline cases with a range of aerosol injection strategies by varying the injection rate, number of injectors, and the timing of the aerosol injection. Our results show that aerosol dispersal is more efficient under pristine conditions due to a transverse circulation created by the gradients in precipitation rates across the plume track. Furthermore, we see that a substantial enhancement in the cloud radiative effect (CRE) is evident in both systems. In the polluted system, the albedo effect (smaller but more numerous droplets causing brighter clouds at constant liquid water) is the
dominant contributor in the initial two days. The contributions from liquid water path (LWP) and cloud fraction adjustments are important on the third and fourth day, respectively. In the pristine system, cloud fraction adjustments are the dominant contributor to the CRE on all three days, followed by the albedo effect. In both these systems, we see that the SCT is delayed due to the injection of aerosol, and the extent of the delay is proportional to the number of particles injected into the marine boundary layer.

## 1  Introduction

Clouds play an important role in the Earth's energy balance. In particular, marine stratocumulus clouds have a net cooling effect on the planet as they reflect a substantial fraction of the incoming solar radiation (Hartmann and Short, 1980). These overcast cloud decks are typically found in the sub-tropics over the eastern flanks of the ocean where sea surface temperatures are colder (Wood, 2012). As these clouds advect towards the equator, they undergo a transition from overcast stratocumulus
to a shallow-cumulus-topped boundary layer with a much lower cloud fraction (Bretherton, 1992; Wyant et al., 1997). Recent studies have shown that aerosol-precipitation interactions play an important role in regulating the stratocumulus-to-cumulus (SCT) transition (Yamaguchi et al., 2017; Zhou et al., 2017). In this study, we explore the impact of aerosol perturbations (ship emissions, deliberate aerosol injection, etc.) on the SCT in the north-east Pacific region.





To elucidate the mechanisms behind the SCT, several studies including field observations (Bretherton and Pincus, 1995; Bretherton et al., 2019) and modeling (Bretherton, 1992; Krueger et al., 1995; Wyant et al., 1997; Sandu et al., 2008; Sandu and Stevens, 2011; Yamaguchi et al., 2017; Erfani et al., 2022) have been undertaken. Until recently, the accepted theory of SCT was attributed to the advection of the cloud layer over a continuously warming sea surface. The increasing sea surface temperature (SST) enhances the surface latent heat flux (LHF). This increases the LWP, which results in enhanced cloud-top entrainment and short-wave (SW) absorption. This promotes the decoupling of the cloud layer from the surface (Krueger et al., 1995; Bretherton and Wyant, 1997; Wyant et al., 1997). Over time the decoupling gets stronger, which enables the formation of overshooting cumulus clouds that locally couple the cloud layer with the surface layer. The enhanced entrainment from cumulus clouds and a lack of steady supply of water vapor from the surface gradually thins and dissipates the stratocumulus layer. Sandu et al. (2010) explored the SCT in four different ocean basins using multiple reanalysis trajectories and concluded that the transition is similar in all cases. These transitions were typically considered to be a multi-day process, based on numerical simulations using microphysical schemes with a fixed cloud droplet concentration ($N_d$). This lack of interaction between aerosol and cloud droplets significantly reduced the degree to which precipitation can influence the SCT. Using the same modeling framework, Sandu and Stevens (2011) explored the factors influencing SCT. Their analysis showed that the SCT is primarily affected by the increasing SST, and the time scale of the transition is governed by the lower tropospheric stability. However, recent simulations with a prognostic aerosol scheme have shown that the interactions among aerosol concentration ($N_a$), $N_d$, and drizzle play an important role in the SCT (Yamaguchi et al., 2015, 2017). The onset of collision-coalescence triggers weak precipitation, which results in lower $N_d$ and $N_a$. This promotes the growth of cloud droplets to larger sizes, which makes the cloud colloidally unstable. This increases precipitation resulting in further reduction in $N_a$, which further strengthens precipitation. This positive feedback (referred to as runaway precipitation) significantly reduces $N_a$ in the boundary layer. Furthermore, the evaporation of sub-cloud precipitation decouples the cloud layer from the surface, which enhances the cumulus activity, thus resulting in a faster SCT. A similar mechanism was proposed by Paluch and Lenschow (1991).

The aerosol perturbations applied in this study should be considered as a proxy for the emissions from ships and deliberate aerosol injection for marine cloud brightening (MCB). MCB is a proposed climate intervention approach where sub-tropical marine stratocumulus clouds are seeded with sea-spray aerosol particles to enhance their reflectivity (Latham and Smith, 1990). Recent studies based on general circulation models have suggested that MCB has the potential to mitigate the warming effects of anthropogenic greenhouse gas emissions (Rasch et al., 2009; Ahlm et al., 2017; Stjern et al., 2018). However, these models do not represent marine stratocumulus with sufficient fidelity, nor account for aerosol-induced cloud adjustments correctly, leaving questions about their ability to assess the net enhancement in cloud reflectivity.

The susceptibility of the cloud radiative effect (CRE) to an aerosol perturbation has three major contributions: $N_d$, liquid water path (LWP), and cloud fraction ($f_c$). The enhancement in cloud reflectivity in response to an increase in $N_d$, stratified by LWP and $f_c$, is known as the Twomey or albedo effect (Twomey, 1974, 1977). In reality, LWP and $f_c$ are affected by aerosol perturbations. The addition of aerosol enhances the colloidal stability of the cloud layer and suppresses precipitation, which increases LWP and $f_c$ (Albrecht, 1989; Goren and Rosenfeld, 2014). However, it also increases the cloud-top entrainment rate through the evaporation-entrainment feedback (Wang et al., 2003) and sedimentation-entrainment feedback (Ackerman et al.,





2009; Bretherton et al., 2007) potentially causing a decrease in LWP and $f_c$. Additionally, LWP and $f_c$ adjustments are affected

by aerosol-enhanced SW absorption (Prabhakaran et al., 2023), and surface flux changes (Chun et al., 2022a).

In this study, we use large-eddy simulations (LESs) to assess the impact of aerosol perturbations on SCT by varying the injection rates and the frequency of perturbations. We consider two SCT baseline systems: polluted (150 particles mg$^{-1}$) and pristine (50 particles mg$^{-1}$). The simulations can be considered of interest to both possible future MCB activities as well as to the broader problem of aerosol-cloud-climate forcing. In the next section, we will present the details of the simulation

setup, including the aerosol forcing function. This is followed by a presentation of simulation results. We end the article with a discussion of the results in the context of MCB, followed by a summary and outlook.

## 2   Methodology

Traditional LES is capable of representing aerosol-cloud interactions faithfully over a wide range of meteorological conditions. However, these studies are limited to rather small domains ($\leq$100 km). Consequently, large-scale meteorological feedbacks

are not captured in such studies. To date, in the context of MCB, most LES studies have been based on fixed meteorological conditions and short durations (12-36 hr) (Wang et al., 2011; Jenkins et al., 2013; Possner et al., 2018; Chun et al., 2022a; Prabhakaran et al., 2023). To understand the impact of MCB-like aerosol perturbations on the SCT, we require domains with horizontal extent spanning several hundred kilometers and time-integration up to three or more days, which are computationally prohibitive at LES resolutions. A good compromise in this regard is Lagrangian LES where a smaller domain with horizontally

uniform properties is advected along the mean wind (Krueger et al., 1995; Sandu et al., 2010). Thus, the spatial variation in the large-scale forcings is represented as temporally varying boundary conditions. Further temporal changes are imposed on the model domain through nudging to a predefined value obtained from coarser models or reanalysis data sets. This methodology has been used to investigate SCT in several studies (Sandu et al., 2010; Yamaguchi et al., 2015; Zhou et al., 2017; Goren et al., 2019), and is used in the current study as well.

The simulations reported here follow the setup in Yamaguchi et al. (2017), and therefore only a brief overview is provided here. The Lagrangian LES model is coupled to a two-moment, bin-emulating, bulk microphysical model (Feingold et al., 1998). The conditions and trajectories are based on the reference Lagrangian SCT case-study developed by Sandu and Stevens (2011). The model domain is advected along the mean boundary layer wind in the north-east Pacific (NEP) region. (See Fig. 1 for a schematic of the trajectory.) The subsidence rates along the trajectory are obtained from Bretherton and Blossey (2014)

and the time evolution of the SST is obtained from Sandu and Stevens (2011).

We use the System for Atmospheric (SAM) model as the LES dynamical core (Khairoutdinov and Randall, 2003). The radiative effects are represented using the rapid radiative transfer model for global climate systems (RRTMG) with extended vertical profiles above domain top (Mlawer et al., 1997). The microphysical scheme consists of two modes representing cloud droplets and raindrops separately. These are represented as log-normal distributions, each with a fixed geometric standard

deviation of 1.2 (Yamaguchi et al., 2017). The two modes are separated by a threshold value of 25 $\mu$m in radius. Additionally, a separate prognostic equation is solved for $N_a$, which includes a fixed surface flux of 70 cm$^{-2}$s$^{-1}$ (Kazil et al., 2011), and

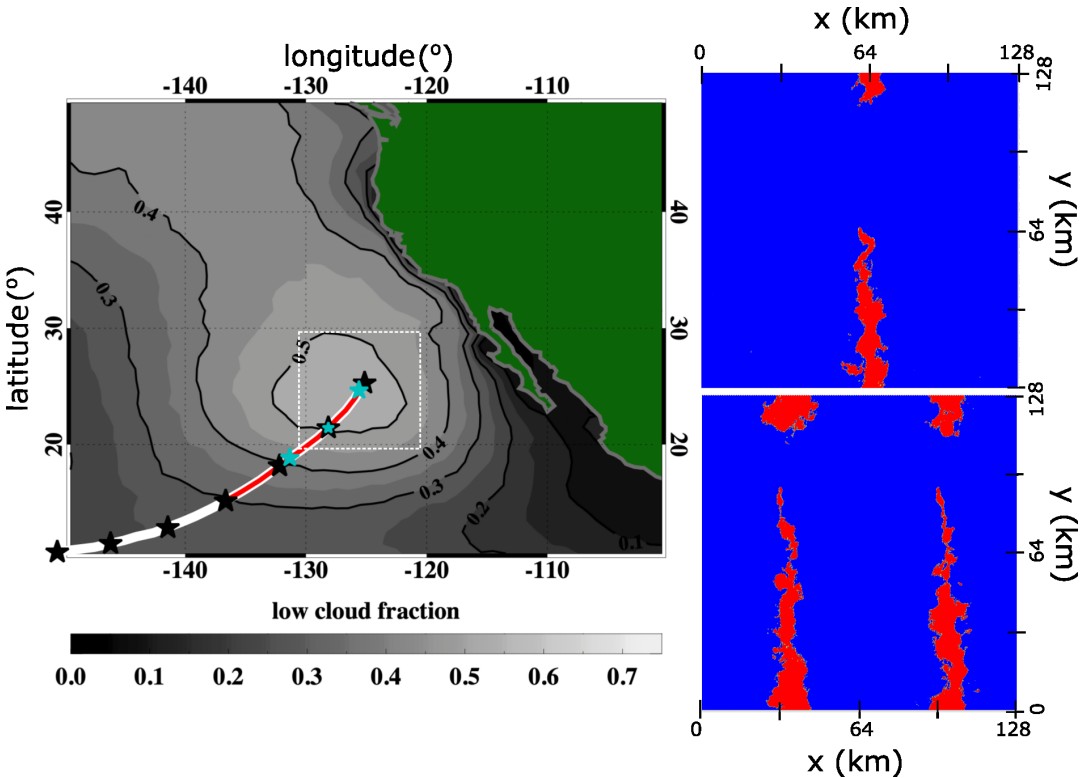

**Figure 1.** The white curve is the 6-day Lagrangian trajectory identified by Sandu et al. (2010) in the NEP. The red curve is the 3-day Lagrangian trajectory simulated here [cf. Sandu and Stevens (2011); Yamaguchi et al. (2017)]. The contours represent the marine low cloud fraction obtained from Aqua-MODIS between 2005 and 2014. The white dashed square box indicates the region studied by Klein and Hartmann (1993). The black stars indicate the air parcel position at 24 h intervals and the cyan stars are the positions of each aerosol pulse. Each aerosol pulse may have one or two active sprayers. The panels to the right represent the one and two sprayer configurations. The red (blue) area in these two panels represent the plume track (background) in the NA150 case identified using the methodology described in the Appendix.

losses or gains through cloud processing (activation, deactivation, collision-coalescence, and wet-removal). The activation of aerosol particles is determined by the local supersaturation, which is calculated prognostically following the semi-analytical method of Clark (1973). The aerosol follows a log-normal size distribution with a geometric standard deviation of 1.5 and a geometric mean radius of 100 nm (Yamaguchi et al., 2017). In the applied modeling framework, cloud processing of aerosol affects the number concentration of aerosol but not the shape of the distribution. Note that the recommended radius of aerosol particles for MCB is thought to be between 15 and 85 nm (Wood, 2021; Haywood et al., 2023), which is slightly smaller than the size range considered. We assume the injected particle size distribution to be the same as the background size distribution to avoid treating two separate populations, which would in any case become indistinguishable once processed by the cloud. This differs from the more rigorous aerosol treatment using the superdroplet approach (Hoffmann and Feingold, 2021; Prabhakaran





et al., 2023; Hoffmann and Feingold, 2023), which is computationally unfeasible for the long simulations and large domains used here. In spite of this simpler aerosol and cloud microphysical treatment, the results are highly relevant in terms of the injection-related modification to $N_d$ and the subsequent adjustments of LWP and $f_c$, which together determine the degree of cloud brightening.

All the simulations have a domain size of 128 km in the horizontal directions and 4.25 km in the vertical direction. A uniform grid spacing of 100 m is used in the horizontal directions. In the vertical direction, a uniform grid spacing of 10 m is used below 2.775 km, and above this height, the grid is smoothly stretched to the domain top with the grid spacing increasing linearly with height. The total number of grid points in the vertical direction is 300. Such a large horizontal domain is chosen to capture the spread rates of the injected aerosol plume, as well as precipitation and associated cloud-field organization (Wang and Feingold,

2009b; Yamaguchi et al., 2017). The time step is set to 3 s with adaptive sub-stepping to satisfy the Courant-Friedrichs-Lewy stability condition. The radiative heating profiles are updated every minute. The simulation is integrated in time for three days, starting on the 196.75th day of the year (July 15, 10 am local time).

Two sets of simulations are conducted for different baseline $N_a$: 150 mg$^{-1}$ (NA150) and 50 mg$^{-1}$ (NA50). These systems are subjected to various aerosol seeding strategies summarized in Tables 1 and 2. We vary aerosol injection rates: low ($1 \times 10^{16}$

particles s$^{-1}$ referred to as 1x) and high ($5 \times 10^{16}$ s$^{-1}$ or $8.6 \times 10^{16}$ s$^{-1}$ referred to as 5x and 8.6x, respectively), which are the recommended ranges per sprayer for MCB (Wood, 2021). Note that the 8.6x injection is explored only for the NA50 system. We also vary the number of aerosol sprayers and the number of aerosol pulses along the trajectory. A schematic of the trajectory, the position of the aerosol pulses, and the configuration of the sprayers is provided in Fig. 1. Each seeding strategy has a five-character code (e.g., 1x-120). The two characters before the hyphen represent the strength of the aerosol injection

rate (0x/1x/5x/8.6x) and the last three digits represent the number of sprayers (0/1/2) active during each aerosol pulse. A value of 0 indicates that no aerosol is injected during the time period of that pulse. The first aerosol pulse is introduced 4 h after the start of the simulation. The next pulse is introduced approximately 20 h after the first pulse and the final pulse is introduced 19 h later. An approximately 20 h separation between pulses is maintained to ensure sufficient time for the aerosol plume to spread across the domain. During this time the cloud layer advects approximately 350 to 400 km. The total aerosol injected

into the marine boundary layer can be calculated as the aerosol injection strength times the sum of the last three digits in the code. Each aerosol pulse represents the passage of sprayer(s) upstream from one end of the domain to the other at a speed of 5 m s$^{-1}$. Each sprayer has the dimension of one grid cell ($100 \times 100$ m$^2$) at the surface.

## 3   Results

### 3.1   NA150: Polluted system

Figure 2a shows the time evolution of the injected aerosol plume areal coverage. The plume is distinguished from the background by setting a threshold on the vertically integrated boundary-layer aerosol concentration ($\overline{N_a}$). A plume is identified when $\overline{N_a}$ exceeds the background variability in $N_a$ (see Appendix A for more details). Ideally, for MCB applications one should consider cloud optical thickness ($\tau$) or cloud albedo ($A_{cld}$) for identifying the plume. However, the signal from these





**Figure 2.** Time series of (a) plume area coverage, (b) liquid water path (LWP), (c) cloud droplet concentration ($N_d$), (d) cloud fraction ($f_c$), (e) domain-averaged precipitation flux (rain rate) at cloud base $z_b$, (f) domain-averaged height of the inversion layer ($z_i$) in NA150. $\tau$ is the cloud optical thickness. The legend is shown in panel (e). The downward pointing arrows at the top of each panel represent the time at which the three aerosol pulses start spraying, if active.

quantities is not very strong for the polluted system, especially in the 1x cases. To leading order, the plume coverage increases

linearly with time in all cases. The spread rate is approximately two times faster for the two sprayer configuration compared to the single sprayer configuration, which is expected. The spread rates are qualitatively similar for the 1x and 5x cases, although the 5x cases appear to be spreading at a faster rate. Note that the turbulent kinetic energy (Fig. 3) is similar in all the cases, until the onset of precipitation, which occurs in the morning of day 3 ($\approx 35$ hours after the start of injection). Thus, for a given number of sprayers, the plume spread rates are for the most part not affected by the number of aerosol particles injected.

Therefore, the slower spread rate in the 1x cases is an artefact of the plume detection methodology. In the 1x cases, along the plume edges, the absolute value of $N_a$ becomes comparable to the background value (low signal-to-noise ratio) after some



time due to dilution. This reduces the detected plume area, as is evident from the decrease in plume area coverage in the 1x cases after sunset on day 2.

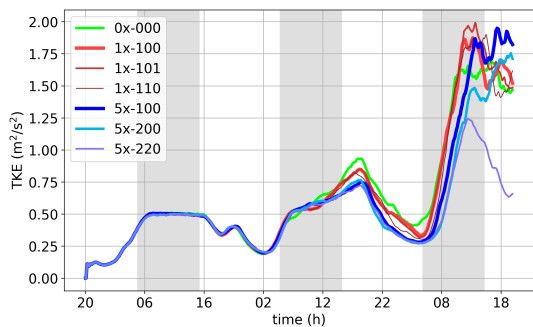

**Figure 3.** Time series of the turbulent kinetic energy (TKE) at 700 m in NA150.

Regarding the plume area fraction, a value of unity is an outcome of the limited horizontal domain. This restricts the scope
of this study in the context of ship tracks because "real" tracks evolve in an "infinite" domain and never reach an area fraction of 1. Consequently, the cloud properties would be continuously affected by spreading and dilution of the track. However, in the context of MCB, the primary focus here, several sprayers are operating in tandem. An area fraction of unity indicates that multiple plume tracks have merged and no more dilution due to spreading is occurring.

Figures 2b, c, and d depict the evolution of the cloud-averaged properties conditional on $\tau > 2$: liquid water path (LWP|
$\tau > 2$), cloud droplet concentration ($N_d \,|\, \tau > 2$), and cloud fraction ($f_c \,|\, \tau > 2$), respectively. Note that we have not separated the plume and background regions when calculating these properties. In the baseline case (0x-000), barring the variability associated with the diurnal cycle, the LWP increases with time in response to the increasing SST and associated marine boundary layer (MBL) deepening (Fig. 2f). This trend continues until day 3 in the afternoon. $N_d$ is nearly constant for the first two days. The reduction in $N_d$ due to weak collision-coalescence is offset by the steady flux of aerosol from the ocean
surface. In the morning hours of day 3, the LWP is high enough to cause precipitation at cloud base ($z_b$) on the order of 0.5 mm d$^{-1}$ (Fig. 2e). On day 3, $N_d$ decreases by about 40% by midday due to (i) collision-coalescence and precipitation losses, and (ii) reduced aerosol activation rate due to the weakening of the updrafts (Fig. 3) due to precipitation evaporation and SW absorption. The subsequent recovery in LWP late in the afternoon triggers runaway precipitation that removes aerosol from the MBL and breaks up the stratocumulus layer. This is evident from the time series of $f_c$, which follows the familiar diurnal
cycle up to day 3 morning. The weak ($<$1 mm d$^{-1}$ at $z_b$) precipitation in the morning and the afternoon enhances the daytime reduction in $f_c$ slightly. However, post sunset, the cloud system recovers and generates sustained stronger precipitation (on the order of 3 mm d$^{-1}$ at $z_b$), eventually reducing $f_c$ to below 30% by the end of the simulation.

$N_d$ increases in all the perturbed cases. After the initial linear increase while the sprayer is active, in the weakly perturbed cases (1x), $N_d$ is nearly constant in time until the morning of day 3. (This is similar to the baseline case.) In the strongly
perturbed cases (5x), there is a steady decrease in $N_d$ with time consistent with the deepening of the MBL (Fig. 2f). The 1x and baseline cases are not affected by this deepening because the difference in $N_a$ between the free troposphere (150 mg$^{-1}$)





and the MBL (150-200 mg$^{-1}$) is not significant, unlike in the 5x cases. Note that in realistic conditions, a strong gradient in $N_a$ can exist between the free troposphere and the MBL. Under those conditions, MBL deepening should be considered as an added factor influencing the evolution of $N_d$ (Yamaguchi et al., 2015). In the LWP time series (Fig. 2b), no significant

changes are evident on day 1 in the perturbed cases as the plume coverage is quite small (Fig. 2a). A weak negative LWP adjustment is visible around midnight on day 2 where the baseline case has the highest LWP and the value in the perturbed cases is lower depending on the total aerosol particles injected into the MBL. The negative LWP adjustment here is an outcome of the entrainment feedback associated with the reduction in the sedimentation flux of droplets (Bretherton et al., 2007) and enhanced evaporation rate near the cloud-top (Wang et al., 2003). During this time, $f_c$ in all the cases is identical.

At sunrise on day 3, in the seeded cases, injection of aerosol suppresses precipitation and increases LWP relative to the baseline case (Fig. 2b,e). The degree of precipitation suppression is proportional to the amount of injected aerosol to that point in time. However, the gain in LWP is not directly proportional to the degree of precipitation suppression, but is partly offset by entrainment effects. Furthermore, the increased $N_d$ sustains slightly higher LWP relative to the baseline case until midday, after which the LWP in the seeded cases decreases below that of the baseline case. Similarly, higher $N_d$ also sustains higher $f_c$

in the morning and lower $f_c$ in the afternoon. This reduction in LWP and $f_c$ is due to enhanced SW absorption associated with the higher LWP and $N_d$ in the morning. The subsequent recovery in LWP and $f_c$ late in the afternoon and early evening triggers strong precipitation in all the cases and significantly depletes $N_a$ and $N_d$ within the MBL. The precipitation ($\approx$ a few mm d$^{-1}$) breaks up the stratocumulus layer as is evident from the decreasing values of $f_c$. The onset of the break-up is controlled by the amount of aerosol injected into the MBL. The delay in the onset of the break-up is proportional to the number of injected

particles into the MBL and is delayed the most in case 5x-220 (Fig. 2e). By the end of the simulation (morning of day 4) weak precipitation has started in case 5x-220. A longer simulation would be required to determine whether this would lead to the break-up of the stratocumulus layer. Additionally, suppression of precipitation deepens the boundary layer, as is evident from the inversion height ($z_i$) on days 3 and 4 (Fig. 2f).

Figure 4a shows the Lagrangian evolution of the vertical profiles of $N_a + N_d$ in case 5x-100, with the cloud-top and cloud-

base heights marked by black lines. Figure 4b shows snapshots of $N_a + N_d$ at intervals of 8 h. Figure 4c shows the vertical integral of sub-cloud negative buoyancy flux integral BFI=$\int \rho_{air} c_p \overline{w'\theta_v'} dz$ ($\forall z < z_b$; $\overline{w'\theta_v'} < 0$), a measure of the decoupling between the cloud layer and surface (Bretherton and Wyant, 1997; Prabhakaran et al., 2023). Here, $\theta_v'$ and $w'$ are fluctuating components of virtual potential temperature and vertical velocity, respectively, and the overline represents the horizontal average. The vertical profiles indicate that the boundary layer is well-mixed until sunrise on day 2, which is supported by the

near zero BFI. Thus, the injected aerosol from the first aerosol pulse mixes throughout the layer. On day 2, the boundary layer is deeper but the vertical mixing is weaker due to the enhanced SW absoprtion from the increase in LWP. This is reflected in the increase in the (negative) magnitude of BFI. The same is evident from the accumulation of aerosol emitted from the ocean surface in the lower levels of the MBL. Post sunset on day 2, the cloud layer continues to deepen, which further strengthens the decoupling from the surface. This continued deepening triggers weak collision-coalescence and precipitation on the morning of

day 3, accompanied by diverging values of BFI (around 02:00). Note that runaway precipitation only occurs after the recovery of LWP and $f_c$ during the night on day 3.



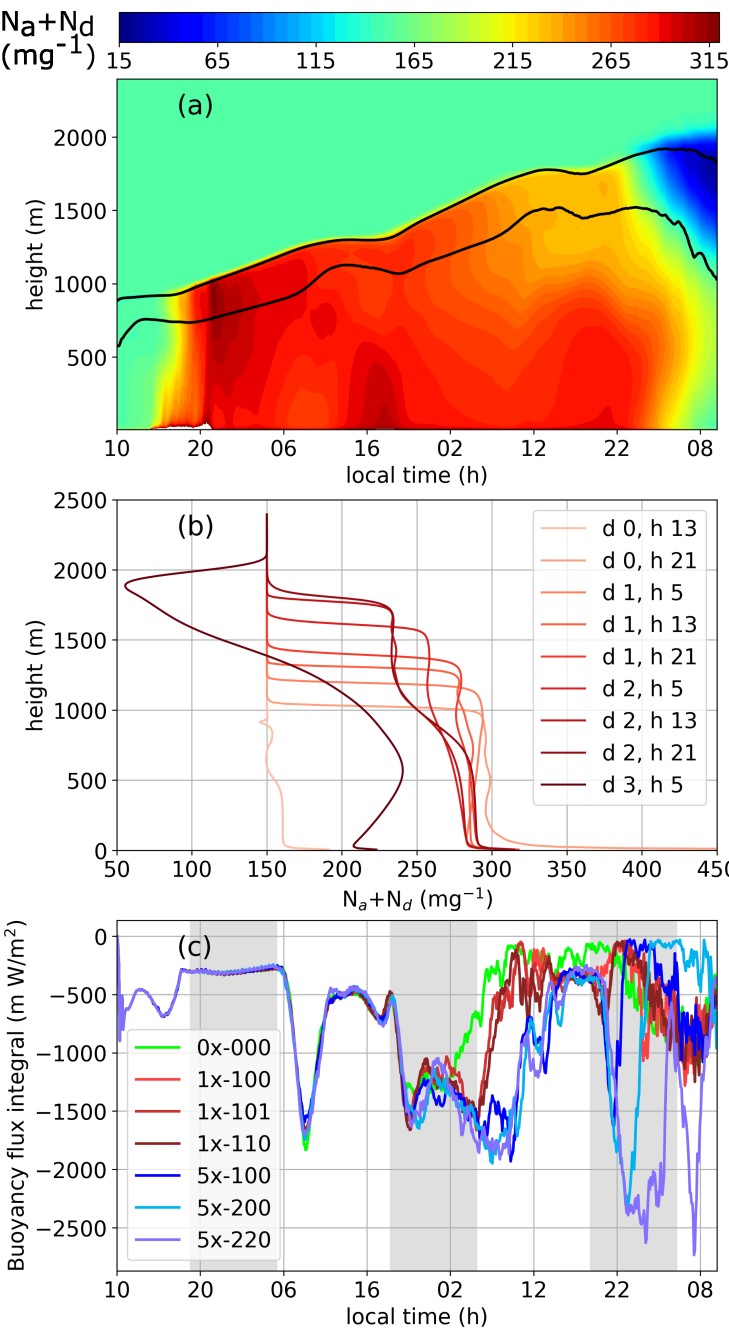

**Figure 4.** (a) Lagrangian curtains of $N_a+N_d$ in case 5x-100, with the black lines representing cloud-base and cloud-top. (b) Vertical profiles of $N_a+N_d$ at select times. (c) Sub-cloud negative buoyancy flux integral, a measure of the degree of MBL mixing for all the cases. More negative values of BFI indicate poorer mixing.



The onset of weak precipitation increases mixing within the sub-cloud layer as indicated by the BFI approaching zero in the 1x cases between 02:00 and 12:00 on day 3. The strong suppression of precipitation in the 5x cases enhances the decoupling due to increased cloud-top entrainment and MBL deepening (see the values of BFI between 02:00 and 10:00 on day 3). For 5x-100, this enhanced decoupling associated with precipitation suppression is also evident from the vertical profiles of $N_a + N_d$ in Fig. 4b (e.g., d 2, h 5). This decoupling is sustained until the onset of runaway precipitation (e.g., d 2, h 13 and d 2, h 21 in Fig. 4b).

**Figure 5.** Time series of changes in CRE relative to the unseeded case and its contributions. (a) dCRE, (b) $N_d$ contribution to dCRE, (c) LWP contribution to dCRE, (d) $f_c$ contribution to dCRE. $\tau$ is the cloud optical thickness. The legend is shown in panel (d).

### 3.1.1 Cloud Radiative Effect

To assess the impact of the various seeding scenarios under polluted conditions, we explore the changes to the cloud radiative effect $\text{CRE} = F_{in}(f_c A_{cld} + [1 - f_c]A_{clr})$, where $F_{in}$ is the incoming solar radiation. Figure 5a shows the changes to the CRE ($\text{dCRE} = \text{CRE}_{0x-000} - \text{CRE}_{\text{case ID}}$) at the top of the atmosphere (TOA) relative to the baseline in all the seeded cases. No





significant changes to CRE are detected on day 1 as the injected aerosol plume track is quite narrow. By day 2, we see a substantial enhancement in the CRE in all the seeded cases. On day 3, the dCRE continues to grow. In some cases, this is due to an increase in aerosol concentration, but even in cases like 1x-100 and 5x-200 where no additional aerosol is added

compared to day 2, we see a greater enhancement in dCRE on day 3. Note that the bulk of the dCRE enhancement occurs in the morning with a dominant peak around 8 to 10 am. A similar result is evident in the simulations of Prabhakaran et al. (2023). In the absence of precipitation, the clouds are thicker in the morning with near 100% cloud coverage. The absorption of incoming SW radiation reduces the LWP and $f_c$ in the afternoon. Consequently, the enhancement in CRE is greatest in the morning. On the afternoon of day 3, there is a hint of cloud darkening. The reason for this is discussed below along with the dCRE

decomposition. On day 4, there is a strong spread in dCRE due to variation in the timing of the transition. In the 1x cases, the dCRE is negligible, whereas the 5x cases with two sprayers show a substantial increase in dCRE. In particular, 5x-220 shows the highest enhancement, with a morning peak value of about $250\,\mathrm{W\,m^{-2}}$, which is well above the peaks on days 2 and 3.

In order to obtain further process level insights, we decompose dCRE to three contributions: $N_d$ ($\mathrm{dCRE}_{N_d}$), LWP ($\mathrm{dCRE}_{\mathrm{LWP}}$), and $f_c$ ($\mathrm{dCRE}_{f_c}$). We follow the procedure laid out in Diamond et al. (2020) and Chun et al. (2022a). The dCRE decomposition

is written as

$$
\begin{aligned}
\mathrm{dCRE} = F_{in}\{ & \underbrace{f_{0\mathrm{x}}[AF_{pl}(A_{cld,N_d,pl} - A_{cld,0\mathrm{x}}) + AF_{bg}(A_{cld,N_d,bg} - A_{cld,0\mathrm{x}})]}_{\mathrm{dCRE}_{Nd}} \\
& + \underbrace{f_{0\mathrm{x}}[AF_{pl}(A_{cld,LWP,pl} - A_{cld,0\mathrm{x}}) + AF_{bg}(A_{cld,LWP,bg} - A_{cld,0\mathrm{x}})]}_{\mathrm{dCRE}_{\mathrm{LWP}}} \\
& + \underbrace{AF_{pl}(f_{pl} - f_{0\mathrm{x}})(A_{cld,pl} - A_{clr}) + AF_{bg}(f_{bg} - f_{0\mathrm{x}})(A_{cld,bg} - A_{clr})}_{\mathrm{dCRE}_{fc}}\},
\end{aligned}
\tag{1}
$$

where $A_{clr}$ is the clear-sky albedo, $A_{cld,N_d/LWP}$ is the cloud albedo contribution from $N_d$ or alternatively LWP, $f$ is cloud fraction, $AF_{pl}$ is plume fraction, 0x, $pl$, and $bg$ in the subscript indicate the baseline case, in-plume track, and off track (background), respectively. Note that $\mathrm{dCRE}_{f_c}$ has contributions from both $f_c$ and $A_{cld}$ adjustments, which is an outcome

of the multiplicative nature of the contributions from $A_{cld}$ and $f_c$ to CRE ($\propto f_c A_{cld}$). For instance, strong changes in $f_c$ are typically associated with precipitation. Under these conditions, LWP and $N_d$, and consequently $A_{cld}$, are not constant. Therefore, $\mathrm{dCRE}_{f_c}$ has contributions from $A_{cld}$ that are not captured in the other two components ($\mathrm{dCRE}_{N_d}$ and $\mathrm{dCRE}_{\mathrm{LWP}}$). The residual from this budget is calculated as $\mathrm{RES} = \mathrm{dCRE} - \mathrm{dCRE}_{N_d} - \mathrm{dCRE}_{\mathrm{LWP}} - \mathrm{dCRE}_{f_c}$. Note that the budget model (Eq. 1) is based on mean-field properties, and does not account for the presence of inhomogeneties within the domain. Thus,

the residual is a measure of the accuracy of the dCRE budget and of the inhomogeneties within the domain (Feingold et al., 2022).

Figures 5b-d show the time series of individual contributions to dCRE from $N_d$, LWP, and $f_c$, respectively. Table 1 provides averaged values between sunrise and sunset for each day. Below, percentage contributions of the three components to dCRE are calculated as $100 \times \mathrm{dCRE}_{N_d/\mathrm{LWP}/f_c}/\mathrm{dCRE}$ using the data from Table 1. The $\mathrm{dCRE}_{N_d}$ component is positive and substantial

on days 2 (over 100%) and 3 (70-100%). In the cases where no additional aerosol is injected after the first pulse, the $\mathrm{dCRE}_{N_d}$



| case ID | day 2 (W m$^{-2}$) | | | | | day 3 (W m$^{-2}$) | | | | | day 4 (W m$^{-2}$) | | | | |
|---|---|---|---|---|---|---|---|---|---|---|---|---|---|---|---|
| | dCRE | $N_d$ | LWP | $f_c$ | RES | dCRE | $N_d$ | LWP | $f_c$ | RES | dCRE | $N_d$ | LWP | $f_c$ | RES |
| 1x-100 | 3.1 | 4.5 | 1.4 | 0.8 | -3.7 | 9.8 | 7.0 | -1.8 | 2.1 | 2.4 | 5.8 | 1.7 | -4.9 | 4.9 | 4.1 |
| 1x-101 | 3.1 | 4.5 | 1.4 | 0.8 | -3.7 | 12.4 | 11.1 | -3.2 | 2.6 | 1.9 | 16.6 | 2.2 | -6.4 | 14.4 | 6.3 |
| 1x-110 | 3.2 | 4.8 | 1.3 | 0.8 | -3.7 | 10.5 | 12.7 | -8.6 | 0.9 | 5.5 | 9.9 | 1.9 | -1.3 | 6.9 | 2.4 |
| 5x-100 | 11.2 | 16.6 | -0.6 | 1.6 | -6.3 | 31.6 | 24.9 | -0.4 | 6.7 | 0.4 | 33.9 | 5.2 | -11.4 | 29.3 | 10.9 |
| 5x-200 | 21.8 | 32.0 | -5.6 | 2.4 | -7.0 | 38.4 | 36.1 | -5.6 | 7.0 | 1.0 | 94.4 | 14.3 | -21.5 | 81.8 | 19.8 |
| 5x-220 | 23.1 | 33.9 | -5.7 | 2.6 | -7.8 | 53.5 | 51.7 | 0.7 | 9.6 | -7.1 | 173.8 | 27.9 | -27.6 | 149.6 | 24.0 |

**Table 1.** Cloud radiative effect enhancement (dCRE = CRE$_{0x-000}$ - CRE$_{\text{case ID}}$) and its decomposition at the TOA for all the seeded cases in NA150. The budgeting is done using cloud properties for a threshold of $\tau > 2$. The quantity for each day is averaged between sunrise and sunset. On day 4, the averaging is done between sunrise and simulation end time. Column RES is the residual of the dCRE budget. RES = dCRE - sum of dCRE components.

component increases from day 2 to day 3 in the single sprayer configuration (e.g., 5x-100 or 1x-100), whereas in the twin sprayer configuration (e.g., 5x-200), the contributions are similar in magnitude with a slightly higher contribution on day 3 because of the suppression of precipitation in the morning. In the single sprayer configuration, apart from the effect of precipitation suppression, the greater areal coverage of the plume on day 3 results in a higher dCRE$_{N_d}$ component. On day 4, the dCRE$_{N_d}$ is less than 20% of dCRE. The dCRE$_{\text{LWP}}$ component is much lower than that of dCRE$_{N_d}$ on day 2 but the two are comparable in magnitude on day 3. Additionally, the LWP contribution is positive in the morning and negative in the afternoon. On day 3, the positive contribution is an outcome of precipitation suppression in the morning. Some of these positive contributions are offset by the effects of entrainment, which explains the lack of a consistent trend in the 5x cases. For instance, case 5x-220 has the strongest precipitation suppression (Fig. 2e), but cases 5x-100 and 5x-200 exhibit higher LWP and dCRE$_{\text{LWP}}$ in the morning. However, in the afternoon, the dCRE$_{\text{LWP}}$ is negative with the highest magnitude for cases 5x-100 and 5x-200, due to the negative LWP adjustment from enhanced SW absorption. The higher LWP in these cases in the morning makes them susceptible to SW absorption, as is evident from the weakly negative dCRE in the afternoon. A similar conclusion was obtained in an earlier study (Prabhakaran et al., 2023). On the morning of day 4, the LWP component is negative and comparable in magnitude to the $N_d$ component. Both of their contributions to dCRE are low (<20%). On this day most of the contribution to dCRE is from the changes in $f_c$ due to precipitation suppression.

From day 2 to day 4, the decoupling of the cloud layer from the surface increases, which favors the development of cumulus clouds. Moreover, the onset of precipitation further reduces the homogeneity within the domain. All of these contribute towards an increase in the magnitude of residual from day 2 to day 4 (column RES in Table 1).





## 3.2 NA50: pristine system

Figure 6, analogous to Fig. 2, shows the evolution of cloud and aerosol properties in the system with $N_a = 50\,\mathrm{mg}^{-1}$. The lower initial $N_a$ leads to an early onset of precipitation, even before the introduction of the first aerosol pulse (Fig. 6e), which leads to a very different aerosol plume and MBL evolution. Figure 6a shows the evolution of plume area coverage. Qualitatively, its evolution is similar to the NA150 system with a monotonic increase in time. However, the spread rate in the current system is higher. For instance, case 5x-200 in NA150 attained a plume area fraction of 0.9 on day 2 around 12:00. The same case in

NA50 attains a similar plume area fraction on day 2 by 03:00. A higher spread rate in NA50 is related to the flow patterns in the precipitating and precipitation-suppressed regions, which is discussed further in Sec. 3.2.1 below.

**Figure 6.** As in Fig. 2, but for case NA50.

Figures 6b, c, and d show the cloud properties LWP, $N_d$, and $f_c$, respectively. In the baseline case, $N_d$ and $f_c$ decrease after the onset of precipitation. The (cloudy-average) LWP shows an increasing trend, as expected in broken cumulus. The stronger





updrafts associated with the convergence of surface flows form deeper clouds with higher LWP, although at the cost of low $f_c$
(Wang and Feingold, 2009a).

In all the perturbed cases, the impact of the seeding is visible in the $N_d$ time series immediately, and in the LWP and $f_c$ time series after about 20:00 on day 1. Injection of aerosol increases $N_d$ and $f_c$, and lowers the LWP relative to the baseline case for much of the duration of the simulation. These reductions in LWP are attributed to the deepening of the MBL (Fig. 6f), and manifest more strongly because of the reduction in $f_c$. After the initial increases in $N_d$ while the sprayers are active, there is
a strong decline (especially in the strong perturbation cases). This decline continues until the aerosol plume spreads across the domain (approximately the middle of day 2). This reduction is due to the ongoing collision-coalescence along the plume track boundaries.

In the 1x cases, the rain rate is slightly lower than the baseline for a few hours post injection (day 1, approximately 20:00). This decrease in precipitation is proportional to the number of injected aerosol particles. During this time the local (plume
track) cloud coverage approaches 100%. By the end of day 1, precipitation in the 1x cases recovers, and exceeds the baseline value, while $f_c$ decreases subsequently, albeit at a slower rate than the baseline case. The higher domain-mean precipitation rate relative to the baseline case is sustained by the generation of non- or weakly precipitating clouds with lower LWPs and higher $N_d$ occurring over a larger fraction of the domain (higher $f_c$). Over time, as the aerosol plume spreads, $N_d$ decreases locally within the plume track (not shown in Fig. 6c). This lowers the colloidal stability of the clouds, resulting in precipitation
and subsequent cloud break-up. By the afternoon of day 2, $f_c$ in the 1x cases is below 0.3 (comparable to the baseline case) with no signs of recovery.

For the higher seeded amounts, the injection of aerosol reduces precipitation significantly for the first 2–3 days (depending on the strength of the injection), allowing the boundary layer to establish a stratocumulus layer with $f_c$ close to 1. Unlike in the 1x cases, the number concentration in the aerosol pulse is high enough to suppress precipitation in adjacent cells due to lateral
spreading. The suppression of precipitation allows the MBL to deepen, which decreases $N_d$ as a result of dilution (Fig. 6f). Subsequent precipitation events towards the end of day 3/day 4 result in a runaway effect and cloud break-up, marking the transition to cumulus clouds.

The counteracting effects on $N_d$ of aerosol injections and MBL deepening play out in an interesting manner in cases 5x and 8.6x. Wang et al. (2011) argued that a concentrated injection is more effective than a distributed injection in enhancing the
CRE in the presence of strong precipitation. The 8.6x-100 case can be considered as a more concentrated version of 5x-200. Compared to case 8.6x-100, 5x-200 has more aerosol injected into the MBL, however $N_d$ in these two cases is comparable – in fact, $N_d$ is slightly higher for case 8.6x-100. The difference comes from the depth of the MBL (Fig. 6f). The MBL height in 5x-200 is about 100-200 m greater than in 8.6x-100. To leading order, the increase in $f_c$ and $z_i$ are proportional to the net injected aerosol concentration. Furthermore, the changes in LHF and sensible heat flux (SHF) in response to aerosol perturbations are
broadly consistent with the deepening of the MBL (Fig. 7). The increased entrainment of drier and warmer free-tropospheric (FT) air enhances LHF and reduces SHF.

Figures 8a and b show the Lagrangian evolution of the vertical profiles of $N_a+N_d$ in cases 8.6x-100 and 5x-200. A few snapshots at intervals of 8 h are shown in Figs. 8c and d for further clarity. The earlier and stronger precipitation suppression



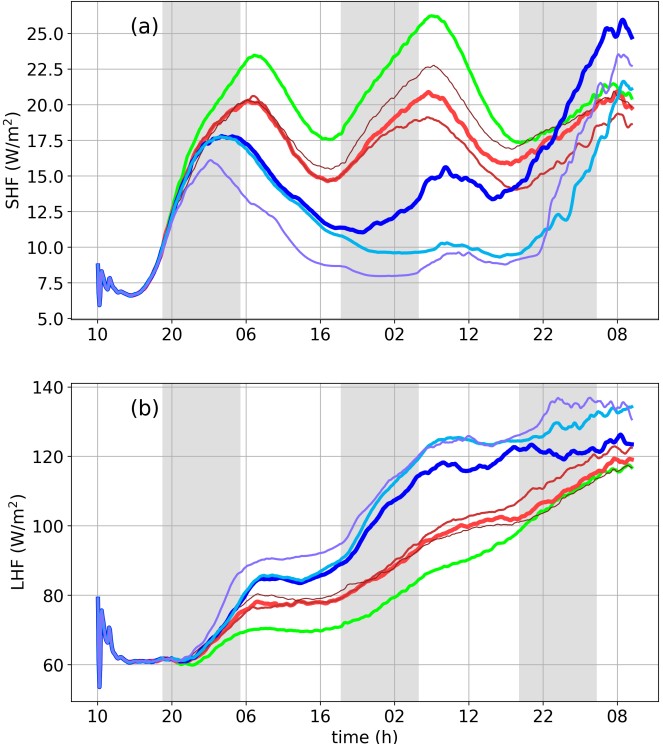

**Figure 7.** Time series of surface scalar fluxes in all the cases in NA50. See Fig. 6 for legend. (a) Sensible heat flux (SHF), and (b) Latent heat flux (LHF).

in 5x-200 starts deepening the MBL by the morning of day 2, which strengthens the decoupling of the cloud layer from the
surface. This is evident from the time series of the BFI (Fig. 9). Furthermore, the deepening of the MBL enhances the LHF
and reduces the SHF (Fig. 7). The corresponding increase in LWP is higher in 5x-200 compared to 8.6x-100, which is evident
on the afternoon of day 2 (Fig. 6b). Furthermore, the dilution associated with the deepening of the cloud layer and weaker net
aerosol vertical transport from the surface layer maintains a lower $N_d$ in 5x-200. Note that $N_a$ is higher in case 5x-200 close
to the surface. Thus, the differences in the final cloud break-up time in these cases is a manifestation of the differences in the
boundary layer structure post aerosol injection.

### 3.2.1 Transverse Circulation

The faster dispersal of the injected aerosol plume in NA50 is associated with the formation of a transverse circulation across
the aerosol plume track (Wang and Feingold, 2009b). The cross sectional snapshots at different times in Fig. 10 show the flow
patterns in the plume affected region and its neighbourhood for case 8.6x-100. Note that the vectors represent the velocity com-
ponent after the mean horizontal wind has been removed. The x-y cross sections in Fig. 10a, c, and e show the organization of
the flow-field from the circulation near the cloud-top along the plume track and its brief evolution in time (about 9 h). This cir-



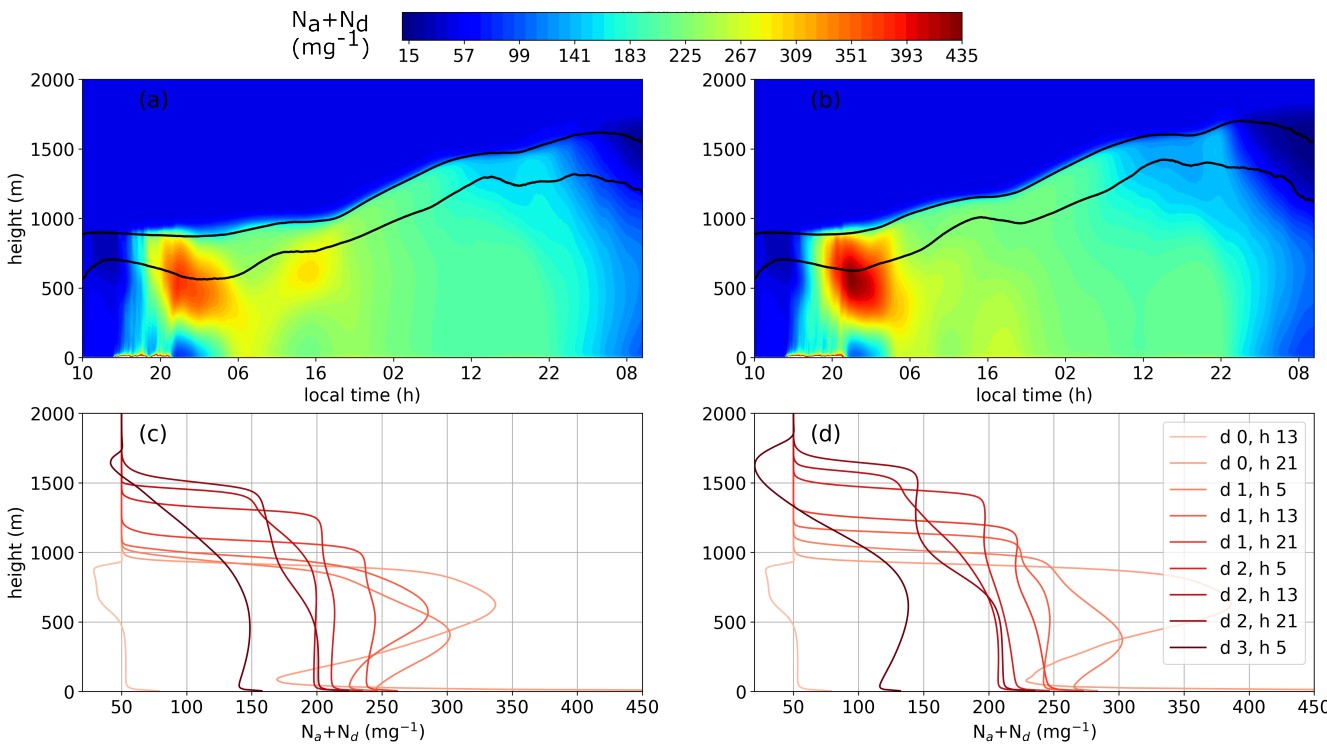

**Figure 8.** NA50. (a) and (b) Lagrangian curtains of $N_a+N_d$ in cases 8.6x-100 and 5x-200 respectively, the black curves represent the cloud-base and cloud-top heights. (c) and (d) Vertical profiles of $N_a+N_d$ at intervals of 8 h in the cases shown in (a) and (b).

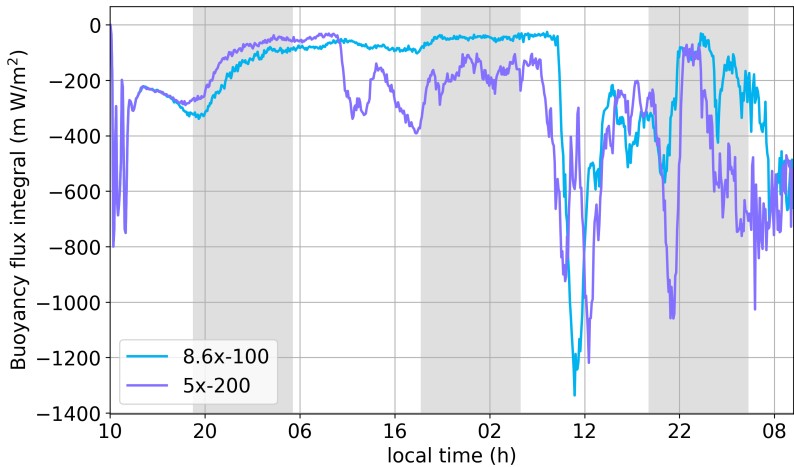

**Figure 9.** Negative buoyancy flux integral in the sub-cloud layer in cases 8.6x-100 and 5x-200.





**Figure 10.** Left panels: x-y cross sections of $N_a + N_d$, right panels: x-z cross sections of buoyancy, for the case 8.6x-100. The vectors represent the planar velocity field after subtracting the horizontal mean wind. x-y cross sections are at z = 700 m and x-z cross sections are at y = 64 km. The green contour lines in the panels to the right indicate liquid water content of value 0.01 g kg$^{-1}$, and black contour lines (dashed) represent values of $N_a + N_d = 100$ and 200 mg$^{-1}$. Top panels: t = day 0, 22 h; middle panels: t = day 1, 1 h; bottom panels: t = day 1, 7 h.




culation is created by the presence of a gradient in the rain rate across the plume track, which causes a corresponding buoyancy gradient that directs the circulation (filled contours in Figs. 10 b, d, and f). Initially, the plume track exhibits slightly positive buoyancy, which is contrasted by strongly negative buoyant regions outside the plume track associated with evaporating precipitation. The strong convergence near the surface along the track is associated with the outflows from the adjacent precipitating cells that supply moisture to the plume track, making it more positively buoyant (Fig. 10b,d). Hence, the injected aerosol particles are lofted to the cloud layer within strong updrafts with velocities around 1-3 $\mathrm{m\,s^{-1}}$ (Fig. 10b, d). This suppresses precipitation, which causes strong outflows near cloud-top at the top of the plume track. These outflows spread horizontally until they encounter a counter flow from a neighbouring cloud cell (x $\approx$ 60 km in Fig. 10d). This deflects the polluted outflow towards the cloud base in the neighbouring cell (Fig. 10d, f), aiding in the faster dispersal of aerosol. Note that the spread rate of the plume is affected by the strength of the perturbation with the 5x cases having a faster spread rate compared to the 1x cases. This is an outcome of the difference in the strength of the transverse circulations. The positive feedback associated with stronger in-track precipitation suppression and subsequent moisture convergence results in a stronger circulation. A detailed discussion about this precipitation gradient-induced mesoscale circulation is provided in Wang and Feingold (2009b) (Fig. 9b therein).

### 3.2.2 Cloud Radiative Effect

Figure 11a shows the time-series of dCRE for the NA50 cases. In all the cases, we see a dominant peak in the morning around 10:00, similar to the dCRE profiles in NA150. In the 1x cases, an enhancement in CRE is evident only on the morning of day 2. In the 5x and 8.6x cases, a substantial enhancement in CRE is evident on days 2 and 3. On day 4, the enhancement in CRE is significant but substantially weaker in comparison to the earlier days. This is due to the precipitation related decrease in $f_c$. Table 2 and Figs. 6b, c, and d show the contributions to dCRE from $N_d$, LWP, and $f_c$, respectively. Note that the y-axis range is different for each panel. As in case N150, the percentage contributions of the three components to dCRE can be calculated from Table 2 as $100 \times \mathrm{dCRE}_{N_d/\mathrm{LWP}/f_c}/dCRE$. The dominant contribution to CRE derives from the changes to $f_c$ in the strong perturbation cases. The contribution from $N_d$ is positive, and its magnitude is less than 20-30% of $\mathrm{dCRE}_{f_c}$. In contrast, the contribution from LWP is negative and is comparable in magnitude to that of $\mathrm{dCRE}_{N_d}$ on days 2 and 3. Similarly, in the 1x cases, the $N_d$ and LWP components are comparable in magnitude but of opposite sign.

## 4 Discussion

In the previous section, we explored the impact of aerosol perturbation on the SCT. We considered two SCT scenarios, one in which strong precipitation only occurs on the afternoon of day 3 (NA150 - polluted), and the other in which strong precipitation occurs on the afternoon of day 1 (NA50 - pristine). The simulation results suggest that an aerosol perturbation delays the onset of the SCT in both scenarios. To leading order, the delay in the transition is proportional to the number of injected aerosol particles in both scenarios, which is broadly consistent with the results of Yamaguchi et al. (2017).





**Figure 11.** Time series of changes to CRE (dCRE) and its contributions for case NA50. (a) dCRE, (b) $N_d$ contribution to dCRE, (c) LWP contribution to dCRE, (d) $f_c$ contribution to dCRE. $\tau$ is the cloud optical thickness. The legend is shown in panel (d). Note that the y-axis range is different for each panel.

In the polluted system, the transition is affected mainly by the total number of injected particles and not by the time sequence of the injections. This is evident from the fact that cases 1x-110 and 1x-101 follow a similar trajectory (except for the time when the plume is still spreading) for all the cloud properties (Fig. 2) and break-up around the same time. A similar conclusion was drawn from the simulations in Prabhakaran et al. (2023) wherein a non-precipitating cloud system was subjected to a range of aerosol perturbations by varying the injection rate and duration of the perturbation. It was concluded that after the initial transient, the cloud system properties were determined only by the total number of aerosol particles injected into the cloud layer. Note that the cloud layer in the polluted simulations qualifies as a non-precipitating system until the morning of day 3, and all the aerosol pulses are active before this time. Furthermore, the addition of aerosol delays this onset of precipitation.

In the pristine system, all the aerosol pulses are active after the onset of precipitation. We see that the distribution in space and time of aerosol pulses plays an important role in the evolution of the cloud system. For instance, 1x-110 and 1x-200 have



| case I.D. | day 2 (W m$^{-2}$) | | | | | day 3 (W m$^{-2}$) | | | | | day 4 (W m$^{-2}$) | | | | |
|---|---|---|---|---|---|---|---|---|---|---|---|---|---|---|---|
| | dCRE | $N_d$ | LWP | $f_c$ | RES | dCRE | $N_d$ | LWP | $f_c$ | RES | dCRE | $N_d$ | LWP | $f_c$ | RES |
| 1x-100 | 15.7 | 10.2 | -8.8 | 11.6 | 2.8 | -2.4 | 8.7 | -3.2 | -4.5 | -3.4 | 2.4 | 2.3 | -0.8 | 1.1 | -0.9 |
| 1x-110 | 14.0 | 12.8 | -8.9 | 11.6 | -1.6 | 0.8 | 17.5 | -10.4 | -3.7 | -2.6 | 5.9 | 4.1 | -1.6 | 3.9 | -0.7 |
| 1x-200 | 19.6 | 10.8 | -9.0 | 17.3 | 0.4 | -2.6 | 4.5 | 0.0 | -4.0 | -3.1 | -0.4 | 2.7 | -0.6 | -1.1 | -1.4 |
| 5x-100 | 53.7 | 30.3 | -21.4 | 42.2 | 2.5 | 78.6 | 21.0 | -18.9 | 73.2 | 3.5 | 12.7 | 1.6 | 6.4 | 8.8 | -4.1 |
| 8.6x-100 | 73.9 | 36.0 | -23.8 | 60.0 | 1.7 | 120.6 | 27.9 | -22.6 | 112.2 | 3.0 | 48.0 | 6.0 | -2.3 | 36.0 | 8.3 |
| 5x-200 | 144.1 | 34.9 | -24.2 | 133.3 | 0.1 | 133.4 | 26.4 | -20.5 | 125.0 | 2.5 | 23.6 | 4.1 | 6.6 | 16.3 | -3.4 |

**Table 2.** As in Table 1, but for case NA50.

the same number of aerosol particles injected into the MBL, however their cloud properties have very different trajectories. Until the afternoon of day 2, $N_d$ and $f_c$ are higher for 1x-200, after which the reverse is true. Note that the timing of this switch-

over is consistent with the injection of the second pulse in 1x-110. The enhancement of precipitation post aerosol perturbation in 1x-200 reduces the injected aerosol concentration within the MBL, however the enhancement in CRE is not significant in either of these cases after day 2. This illustrates the complexity in the evolution of the MBL properties in this system and is reinforced by the fact that the onset of SCT is not delayed the most for the strongest perturbation (5x-200), but by a slightly weaker perturbation (8.6x-100). The key difference is the depth of the inversion layer, which is proportional to the magnitude

of the precipitation suppression. This enhanced depth dilutes $N_a$ and also strengthens the decoupling of the cloud layer from the surface. The increased LHF due to entrainment deepening and stronger cumulus clouds trigger precipitation that results in the earlier transition in 5x-200 compared to 8.6x-100.

The CRE in both the polluted and pristine systems is enhanced post-aerosol perturbation. No substantial darkening tendency is evident in any of the simulations. The decomposition of CRE sheds insights into the contributions from $N_d$, LWP, and $f_c$. In

the polluted system, the dCRE increases from day 2 to day 4. The contributions from the negative LWP adjustment are around 10-30% of $N_d$. On day 3, the positive adjustment in LWP in the morning is due to precipitation suppression, some of which is offset by the entrainment adjustment. The negative LWP adjustment in the afternoon is due to enhanced SW absorption. These counteracting effects reduce the net contribution from the LWP component (see Tab. 1). Note that the negative adjustment in LWP due to entrainment (dominant during the night) is not significant in this system due to fairly high free-tropospheric

humidity ($\approx$3.5 g kg$^{-1}$). On day 4, the enhancement in CRE is largely a result of the changes to $f_c$ associated with precipitation suppression, and its peak magnitude is approximately 75% more than the dCRE peak on day 3. In the pristine system, the SCT is delayed the most in case 8.6x-100, resulting in the highest dCRE on day 4. However, the net brightness is not the highest for this case as the dCRE in case 5x-200 is substantially higher on days 2 and 3. Additionally, the dominant contribution to dCRE is from the changes to $f_c$ associated with precipitation suppression, which is consistent with earlier LES studies on MCB





(Wang et al., 2011; Jenkins et al., 2013; Chun et al., 2022a; Prabhakaran et al., 2023). On day 2, the contribution from $N_d$ is also substantial, and comparable to $f_c$.

## 4.1   CRE vs $N_d$

Using satellite data, Goren et al. (2022) showed recently that in spite of the saturation in cloud brightening that occurs at higher $N_d$ (the albedo effect), CRE increases linearly with increasing $N_d$. This linear relationship is an outcome of the effect of $N_d$ on

$f_c$ via delayed precipitation. This is consistent with our simulations where we see a proportionate delay in SCT with increasing aerosol injection. Figure 12 shows that the daily maximum CRE increases with $N_d$ but is a strong function of the solar zenith angle (SZA). Goren et al. (2022) do not account for the variability in SZA, and use the average insolation. The peak values in CRE on each day occur between 8 and 10 am in all cases. In the NA150 and NA50 cases (top and bottom panels in Fig. 12) we see a linear increase in CRE peaks with $N_d$ on day 2. This is an outcome of the spreading of the plume areal coverage ($AF_{pl}$),

and CRE is directly proportional to $AF_{pl}$ (see Eq. 1). On day 3 in the NA150 cases, we see a hint of saturation in CRE, which could be related to the albedo effect (proportional to $N_d^{1/3}$). On day 4 in NA150, as well as day 3 in NA50, we see a near linear increase in CRE with $N_d$. This is an outcome of precipitation suppression and corresponding changes to $f_c$. However, we see a weak hint of saturation in the NA150 cases on day 4, which could be related to higher $N_d$ in the NA150 cases. Note that the $N_d$ range considered in Goren et al. (2022) is rather small (<100 cm$^{-3}$) compared to the range considered here

(between 25 and 500 cm$^{-3}$). Furthermore, the saturation in CRE could be related to the reduction in LWP from enhanced SW absorption and cloud-top entrainment, effects which were not taken into account in the satellite data analysis of Goren et al. (2022). Currently, we do not have enough data to confirm this hypothesis. Thus, more studies under different meteorological conditions are required to ascertain the nature of the relationship between CRE and $N_d$ in the SCT.

## 4.2   MCB and SCT

These insights indicate that if one considers deliberate injections of aerosol into NEP clouds where precipitation is not imminent (moderately to very polluted clouds), the focus should be to inject as many aerosol particles as possible into the MBL [until coagulation losses start to dominate. cf. Baker and Charlson 1990] to enhance the brightness of the cloud deck. The aerosol perturbation should be performed while the ocean surface temperature is relatively cold as advection towards warmer waters strengthens the decoupling of the cloud layer from the surface, which reduces the efficiency of the vertical transport

of the injected aerosol. Furthermore, since the spread rate of the aerosol plume is low under these conditions, a more widely distributed injection would aid in a faster dispersal of the injected aerosol. In the context of the pristine system, a high aerosol injection rate is required for a successful implementation of MCB. Since the aerosol plume spread rate in this system is high, the number of sprayers required could be lower. Additionally, if targeted brightening is considered, the pristine system would be more effective due to the substantial enhancement in the CRE due to rapid and strong $f_c$ increases soon after aerosol injection

ends. In the polluted system, such strong changes in CRE are evident only after several hours post aerosol perturbation.

A key question that arises from the results and the discussion here is: to what extent can the SCT be delayed through aerosol perturbations? Additionally, if a substantially higher aerosol concentration is injected into the MBL, would that result in the





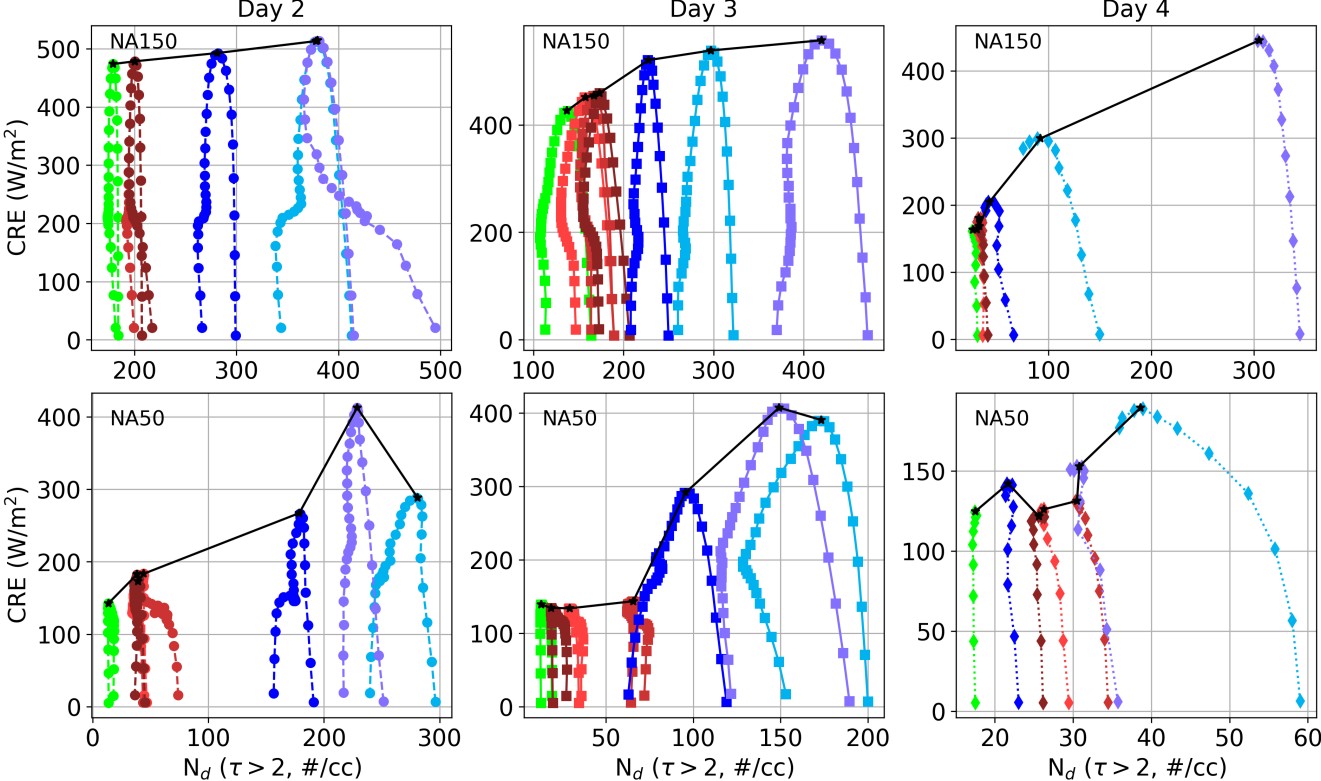

**Figure 12.** CRE vs $N_d$ for each day in NA150 (top panel) and NA50 (bottom panel). The color code for the top and bottom panel figures is the same as Figs. 5 and 11, respectively. The separation between consecutive symbols indicates a time difference of 24 minutes. The black line connects the maximum values in CRE for each case.

classic SCT scenario where the transition is due to the warming of the ocean surface and not due to precipitation? The injection of aerosol enhances the colloidal stability of the cloud layer and suppresses precipitation but it also enhances the entrainment rate of free-tropospheric air, which reduces LWP. However, higher aerosol concentrations deepen the cloud layer, and thus have a tendency to enhance LWP (Stevens and Feingold, 2009). The significance of these competing effects may vary on a case-by-case basis. Therefore, a wider variety of simulations under different conditions are required to address these questions.

## 5   Summary and Outlook

In this study, we explored how the stratocumulus-to-cumulus transition (SCT) is affected by deliberate aerosol perturbations using a Lagrangian large-eddy simulation (LES) model coupled to a two-moment bulk microphysics scheme. We used the average trajectory from Sandu and Stevens (2011). The setup of these simulations is directed towards marine cloud brightening (MCB)– the deliberate injection of aerosol particles into the marine boundary layer to enhance the brightness of the marine



stratocumulus clouds, thereby exerting a cooling effect on the planet. We considered two different baseline aerosol conditions: a polluted (150 particles mg$^{-1}$) background and a pristine background (50 particles mg$^{-1}$). We varied the aerosol injection
rates per sprayer, number of sprayers, and number of aerosol pulses to assess the impact of various MCB strategies. Our results showed that the spread rate of the aerosol plume is faster in the pristine system due to the transverse circulation induced by the gradient in rain rate across the plume track. In response to the aerosol perturbation, the SCT is delayed in both polluted and pristine systems. To leading order, in the polluted scenario, the time delay in the transition is proportional to the amount of aerosol injected into the MBL and is only weakly affected by the distribution (in space and time) of the aerosol sprayers.
The enhancement in cloud radiative effect (CRE) increases from day 2 to day 4. The changes in CRE are dominated by the albedo effect on days 2 and 3, and cloud fraction ($f_c$) adjustments on day 4. In the pristine system, only the strong perturbations make a substantial contribution to MCB. The weak perturbations are dissipated within a day through enhanced precipitation in the aerosol plume track. The time delay in SCT is affected by the total number of aerosol particles injected into the marine boundary layer and their distribution in space and time. A more concentrated but slightly weaker aerosol injection tends to
delay the SCT more effectively than splitting it across two sprayers. The enhancement in CRE is dominated by $f_c$ and is sustained for two days in the strongly perturbed cases.

The results presented here are based on the composite trajectory from a five-year (2002-2007, May-October) climatology. This average trajectory may mask the variability in profiles, which may impact the SCT. For instance, faster advection or a faster SST increase may result in an early transition. In such a scenario, the time required for the aerosol plume to spread
and the corresponding cloud adjustment time scales may affect the effectiveness of MCB. In other words, a different aerosol injection rate and sprayer configuration may be required under these conditions for the effective implementation of MCB. Thus, future studies should use more realistic conditions based on instantaneous soundings and forcings. Additionally, the current Lagrangian model does not account for the large-scale feedback associated with aerosol perturbation. Thus, further model improvements are warranted to better constrain the impact of aerosol perturbation (Chun et al., 2022b).

*Code and data availability.* The simulations were carried out using SAM (https://wiki.harvard.edu/confluence/display/climatemodeling/SAM). The data from the simulations is available here https://csl.noaa.gov/groups/csl9/datasets/data/cloud_phys/2023-Prabhakaran-etal/.

**Appendix A: Aerosol Plume Identification**

The aerosol plume is detected by setting a threshold manually on the vertically integrated ($0 \leq z \leq z_i$) aerosol concentration. For each of the baseline systems (NA150 and NA50), the same threshold values are used in all the perturbed cases. In the
NA150 case, the background aerosol concentration has a variability of $\pm 10\%$ about the mean value. We use a spatial Gaussian filter of five pixels width to smooth these fluctuations. This causes artificial broadening of the plume during its initial evolution period. However, with time, the effect of the filter is weakened due to the increase in plume area coverage. In the NA50 system, the signal-to-noise ratio is very high even for the weak perturbations (1x). Thus, no filtering is used in determining the plume



area coverage in NA50. The time series of the threshold values used in this study is shown in Fig. A1. The threshold values
change with time due to the changes in the background conditions. In the NA150 system, the threshold value is nearly constant,
with minor (within 15% of the start value) changes during the evolution of the system. On the other hand, in the NA50 system,
the threshold value decreases quickly with with time to account for the losses from precipitation in the region away from the
plume track. Since the threshold values are the same for all the aerosol plume tracks in each system, a sensitivity test on these
threshold is not required as the relative trend in the spread rates would be similar.

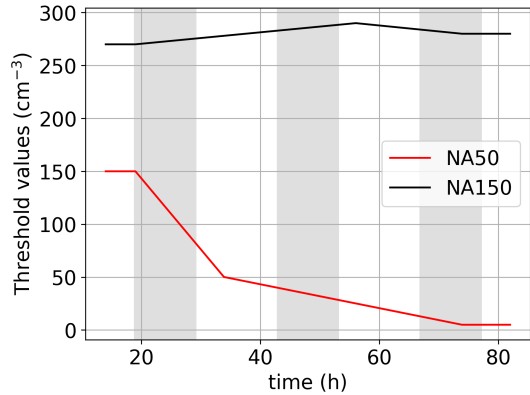

**Figure A1.** Time series of the threshold values for identifying plume in NA150 and NA50.

*Author contributions.* PP, FH, and GF designed the research. PP carried out the simulations and analysis. PP, FH, and GF discussed the
results. PP wrote the manuscript with input from FH and GF.

*Competing interests.* At least one of the (co-)authors is a member of the editorial board of *Atmospheric Chemistry and Physics*. The peer-
review process was guided by an independent editor, and the authors also have no other competing interests to declare.

*Acknowledgements.* This research was supported by the U.S. Department of Commerce, Earth's Radiation Budget grant, NOAA CPO Cli-
mate and CI no. 03-01-07-001. FH appreciates support from the Emmy Noether program of the German Research Foundation (DFG) under
grant HO 6588/1-1. Marat Khairoutdinov graciously provided the SAM model. Dr. Jianhao Zhang, CIRES/NOAA-CSL, provided the data
for Fig. 1.





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
