# Peer review of "Effects of Intermittent Aerosol Forcing on the Stratocumulus-to-Cumulus Transition"

_EGUsphere, 2023_

## Referee Comment (RC1)

Review of „Effects of Intermittent Aerosol Forcing on the Stratocumulus-to-Cumulus Transition"
by Prabhakaran et al.

This paper assesses how conceivable MCB strategies may influence stratocumulus-to-cumulus transitions (SCT). Different seeding strategies are explored here. The study is scientifically sound and includes findings of great interest to the community. My main concern with respect to their methodology is the use of a climatological forcing, which linearises the system to an extent, that may not be valid. However, this limitation is clearly acknowledged by the authors. Remaining concerns listed below are with respect to the interpretation of their Na=50mg-1 experiments, and the occasional lack of transparency of presented arguments.
However, overall it is a well-written manuscript and I recomment publication with minor revisions.

General:

You chose your experiment with a fixed seeding geometry with respect to the domain. However, this means that the orientation of the seeding source will constantly change with respect to the wind direction inside the domain, right? Is there any impact of large-scale advection on top of the mesoscale organisation driving lateral mixing of aerosol?

Section 3.2: I would like the authors to add more context ot this set of experiments. Currently, the results are presented as investigating the sensitivity of an SCT under lower background aerosol concentrations. This is also consistent with the manuscript title. However, Fig. 6 suggests, that you are looking at this in a quite different system to start with. The transition seems to have already occurred before the analysis and indeed the first aerosol injection. Was there even a transition prior to hour 10? I am also not sure how good an analogue it is for a post-transition type of experiment, since the simulated boundary layer you start with is quite shallow. This is also neccessary for context of interpretation. How often do such shallow cumulus layers occur so close to the coast and thus how relevant is it for understanding the effect of potential MCB applications?

Section 3.2.1: Following on from my previous comment. While I scientifically do not object to the analysis and mechansim presented, it is shown in a boundary layer, whichs representativeness is questionable (previous comment). Would you expect to see the same in deeper cumulus-topped BLs as well? Once again, this is pointing towards relevance and generalisability of your finding. Also, please clarify in the text the novelty/added value of your findings to the previous mechanism presented in Wang & Feingold 2009b?

Please explain why different sprayer configurations and specificiations were performed in Na=150mg-1 and Na=50mg-1 experiments. Why was 8.6x-100 introduced but 5x-220 omitted?

Specific:

L25ff: I don't disagree with the reviewers that precipitation may play a role in accelerating the Stratocumulus-cumulus transition. Howevever, observation-based assessments also support the notion that stratocumululs-cumulus transitions are predominantly driven by entrainment rather than precipitation (e.g. Bretherton et al. 2019, Eastman et al 2021). It seems misleading to attribute these differences simply to the neglect of aerosol-cloud interactions in numerical models.

Reference: Eastman, R., I. L. McCoy, and R. Wood, 2021: Environmental and Internal Controls on Lagrangian Transitions from Closed Cell Mesoscale Cellular Convection over Subtropical Oceans. *J. Atmos. Sci.*, **78**, 2367–2383, https://doi.org/10.1175/JAS-D-20-0277.1.
Bretherton, C. S., and Coauthors, 2019: Cloud, Aerosol, and Boundary Layer Structure across the Northeast Pacific Stratocumulus–Cumulus Transition as Observed during CSET. *Mon. Wea. Rev.*, **147**, 2083–2103, https://doi.org/10.1175/MWR-D-18-0281.1.

L120: suggest rephrase „during each pulse" -> „during each of the 3 pulses". Or something like this. It took me a minute to understand why there are 3 numbers behind the hyphon.

L127: At which altitude to you spray?

L137: Please explain why the TKE is used here as a metric for lateral dispersion. I would have thought that the length-scale of mesoscale organisation may be a better constraint?

L145ff: I agree with the authors on this. The fact that you reach an area fraction of 1 suggests, that your results in the 5x scenarios are likely limited by domain size. This reduction of the analysis to ship track scales, should be made more clearly in the abstract. The current first sentence is ambiguous in this regard. This holds especially for all conclusions based on the $Na=50mg-1$ background experiments discussed in the following section.

L157: „weakening of the updrafts (Fig. 3). Where do I see this in Fig. 3? What is included in your metric of TKE (i.e. subgrid and grid-scale components)

L166ff: Why is the deepening during the first days a function of the difference in Na across the inversion in your simulations? This is not clear. Is it not rather that incloud Nd is not increased that much in the 1x experiments, such that entrainment feedbacks are not effective?

L175ff: Your explanations of the observed behaviour in LWP and fc in seeded experiments lack evidence. Please provide this (at least as part of this review).

L184: I agree with your statement, but please define the „onset of the transition".

Fig4: Please add injection times like in Fig. 2. Hours 5, 13, 21 are unconventional. Why were they chosen? Also maybe you could add a timeline for shortwave absorption here to also provide partial proof for your arguments presented earlier (i.e. comment on L175ff).
Your time labels in Fig. 4b (i.e. starting at day 0) are different to the rest of the paper. Please adjust. Generally, each figure caption showing temporal slices with a periodic time axis should state the starting day for clarity. I also believe the day0 label in Fig.4b refers to the period marked under „day 1" in Figure 2 and throughout the manuscript.

L189ff: I am not sure that your metric of BFI is really so usefull to diagnose decoupling in this case. For instance you simulate low BFI in periods following day 2 h12 (as low in the first 24 coupled hours) while the profile of Na+Nd clearly indicate decoupling. Other metrics have previously been suggested in the literature that may show this more clearly?

Fig5: The shift im time-axis from day-to-day (i.e. first 10, 20, 6h,16,02,...) makes it hard to look at the periodicity of the cloud-radiative effect. I would suggest a uniform periodic time axis.

L288: „changes in fc and are proportional to net injected aerosol concentration" The term „net injected" I find misleading. There is no sink in an injection. I would rephrase. Also, where do I see this?

Fig. 8. Same questions and comments apply as in Fig. 4.

L348: rephrase „affected mainly by the total number of injected particles" to „ affected mainly by the total number of injected particles prior to the transition". I beleive your statement only holds under these conditions, as only then the number of injected particles is the same. Otherwise, this statement cannot be true, as temporarily the number of injected aerosol will by design be different between 1x-101 and 1x110.

L367: „earlier transition" what transition? It already starts out in the cumulus state in 0x-000 right? I think the suggested definition of the transition in an earlier comment can help clarify this point.

L391: According to table 1 there is no saturation in the Twomey effect though. There is a ~20 W/m2 difference between 5x-200 and 5x-220 on day 3.

---

## Referee Comment (RC2)

Review comment on "Effects of Intermittent Aerosol Forcing on the Stratocumulus-to-Cumulus Transition" by Prabhakaran et al.,

General comment:

The authors investigated the impact of injected aerosols on the transition from stratocumulus to cumulus clouds in the north-east Pacific. They revealed that aerosol injection delays the stratocumulus-to-cumulus transition, with the extent of the delay directly proportional to the number of aerosol particles injected into the marine boundary layer, ultimately influencing cloud radiative effects in both pristine and polluted systems. Overall, I suggest publishing it after addressing the comments below.

Major comments:

- Injections time are chosen to be the same stage during diurnal cycle. Why is that? How will it impact the results if injection at different time during the diurnal cycle?

Minor comments:

- Line 75: why? the deficiency with small LES domain is the lack of feedback from larger-scale, but the Lagrangian LES does not solve this problem, even the large-scale variabilities are presented by forcings. On top of that, whether forcing is good enough is a new problem.
- Line 82: recommend 1-2 sentence to clarify what does "bin-emulating, bulk microphysical model" mean.
- Line 90: "The two modes are separated by a threshold value of 25 .m in radius." reference? It is larger than the convectional 12 or 13 micro as the start of auto-conversion. Why?
- Line 96: "In the applied modeling framework, cloud processing of aerosol affects the number concentration of aerosol but not the shape of the distribution." Reference?

- Line 102-104: "the results are highly relevant in terms of the injection-related modification to and the subsequent adjustments of LWP and fc, which together determine the degree of Nd cloud brightening." Besides LWP and fc, how about cloud base height, cloud top height and cloud depth?

- Line 130: "the plume", what does the "plume" mean? Injected aerosol plume? Or cloud plume? I don't think I understand why the authors can use aerosol concentration to represent a plume, that usually use cloud optical thickness or cloud albedo to identify. They are two quite distinguished concept. Also, does it make more sense to use na+nd to identify plume than solely na?

- Line 135-137: which figure are you referring to representing spread rate?

- Line 151-152: why? Why not separate the plume and background region?

- Line 156-158: "On day 3, decreases by about 40% by midday due to (i) collision-coalescence and precipitation losses, Nd and (ii) reduced aerosol activation rate due to the weakening of the updrafts (Fig. 3) due to precipitation evaporation and SW absorption." Figure 3 does not updraft. Also, how do you know the activation fraction is reduced? If that is inferred, soften the sentence to reflect that.

- Line 164: remove the bracket

- Many places explain the phenomena by entrainment but there is no direct reflection of the cloud top entrainment strength. How about adding it to the figures?

---

## Author Comment (AC1)

**Response to Reviews of "Effects of Intermittent Aerosol Forcing on the Stratocumulus-to-Cumulus Transition"** - Prabhakaran *et al.* 2023

**Reviewer 1:**

Summary:

This paper assesses how conceivable MCB strategies may influence stratocumulus-to-cumulus transitions (SCT). Different seeding strategies are explored here. The study is scientifically sound and includes findings of great interest to the community. My main concern with respect to their methodology is the use of a climatological forcing, which linearises the system to an extent, that may not be valid. However, this limitation is clearly acknowledged by the authors. Remaining concerns listed below are with respect to the interpretation of their Na=50mg-1 experiments, and the occasional lack of transparency of presented arguments. However, overall it is a well-written manuscript and I recomment publication with minor revisions.

We thank the reviewer for the positive evaluation of the paper and the constructive comments. In our response below, the reviewer comments are in black and our responses are in blue.

General:

You chose your experiment with a fixed seeding geometry with respect to the domain. However, this means that the orientation of the seeding source will constantly change with respect to the wind direction inside the domain, right? Is there any impact of large-scale advection on top of the mesoscale organisation driving lateral mixing of aerosol?

We thank the reviewer for this question. The seeding geometry is not fixed in our study. The relative speed between the sprayer and the mean horizontal wind is 5 m/s. We have stated this more clearly in Lines 145-146 in the revised manuscript. The effect of large-scale advection is relevant locally while the sprayer is active. In each grid box the sprayer is active for 20 seconds (100 m/ 5 ms-1). While the sprayer is inactive, the injected aerosol particles are only advected by the large-scale flow with no enhanced lateral mixing from larger scales. Furthermore, the trajectory of the sprayer is closely aligned with the mean wind. Thus, the impact of the large-scale advection in the lateral spreading of the aerosol plume is rather weak.

Section 3.2: I would like the authors to add more context ot this set of experiments. Currently, the results are presented as investigating the sensitivity of an SCT under lower background aerosol concentrations. This is also consistent with the manuscript title. However, Fig. 6 suggests, that you are looking at this in a quite different system to start with. The transition seems to have already occurred before the analysis and indeed the first aerosol injection. Was there even a transition prior to hour 10? I am also not sure how good an analogue it is for a post-transition type of experiment, since the simulated boundary layer you start with is quite shallow. This is also neccessary for context of interpretation. How often do such shallow cumulus layers occur so close to the coast and thus how relevant is it for understanding the effect of potential MCB applications?

We thank the reviewer for this question. The simulations presented here are fairly idealized using climatological forcings. Similar to earlier studies (e.g., Sandu & Stevens 2011, Yamaguchi et al 2017), we have varied the background aerosol concentration to create a pristine and polluted boundary layer. In Fig. 6, our objective was to look at how aerosol perturbation would affect a stratocumulus-topped boundary layer influenced by precipitation. Post precipitation, NA50 cloud system has an open-cellular structure that is quite different from the closed-cellular structure before precipitation. Note that the transition here is driven by precipitation due to a cleaner boundary layer (and not due to the changes in the surface fluxes). Consequently, the boundary layer is quite shallow compared to NA150 and hence this transition is reversible (Feingold et al 2015). We agree with the reviewer that the frequency of occurrence of such cleaner boundary layer is an important aspect to consider in the context of MCB. In the revised draft, we have provided more context for the choice of NA50 and have discussed the

frequency of occurrence of NA50 (Lines 131-135, 289-292, 446-452).

Section 3.2.1: Following on from my previous comment. While I scientifically do not object to the analysis and mechansim presented, it is shown in a boundary layer, whichs representativeness is questionable (previous comment). Would you expect to see the same in deeper cumulus-topped BLs as well? Once again, this is pointing towards relevance and generalisability of your finding. Also, please clarify in the text the novelty/added value of your findings to the previous mechanism presented in Wang and Feingold 2009b?

As was discussed in the previous reply, the boundary layer in NA50 is representative of a pristine boundary layer near the shore. In the revised draft, we have acknowledged that the frequency of occurrence of such cleaner MBL near the coast is quite low. However, with the latest shipping emission regulations and a projected reduction in emissions, a cleaner MBL nearer (relatively) to the coast may become more realistic in the future. We have mentioned this in the discussion. Lines 446-452.

The meteorological and boundary layer conditions in the present study is very different from that in Wang & Feingold 2009b. This supports the generality of the mechanism responsible for the higher lateral spread rate of the aerosol plume in a precipitating system. We have stated this in the revised draft. Lines 359-362.

The question of whether we would see a similar mechanism in a deeper cumulus-topped boundary layer is an interesting one, but outside the scope of the current study. The initial transition in the NA50 system is driven by precipitation. Note that this mechanism responsible for the enhanced spread rate will work in the presence of precipitation in an open-cellular configuration. It is not clear if this mechanism would work in a deeper cumulus-topped boundary layer with precipitation. A different case study is required to address this question and will be part of future work.

Please explain why different sprayer configurations and specifications were performed in Na=150mg-1 and Na=50mg-1 experiments. Why was 8.6x-100 introduced but 5x-220 omitted?
We have carried out 6 perturbed simulations for each of the baseline cases. Each of these simulations require over 100000 core hours. Since we ran the 8.6x-100 case for NA50, the motivation for which is mentioned in the article (Lines 323-327), we chose not to run 5x-220 case due to resource constraints. Note that this does not affect the conclusions drawn from these simulations.

**Specific:**

L25ff: I don't disagree with the reviewers that precipitation may play a role in accelerating the Stratocumulus-cumulus transition. However, observation-based assessments also support the notion that stratocumululs-cumulus transitions are predominantly driven by entrainment rather than precipitation (e.g. Bretherton et al. 2019, Eastman et al 2021). It seems misleading to attribute these differences simply to the neglect of aerosol-cloud interactions in numerical models.
We have rephrased the text in the introduction. Lines: 40-57.

**Reference:**

Eastman, R., I. L. McCoy, and R. Wood, 2021: Environmental and Internal Controls on Lagrangian Transitions from Closed Cell Mesoscale Cellular Convection over Subtropical Oceans.

J. Atmos. Sci., 78, 2367–238. Bretherton, C. S., and Coauthors, 2019: Cloud, Aerosol, and Boundary Layer Structure across the Northeast Pacific Stratocumulus–Cumulus Transition as Observed during CSET. Mon. Wea. Rev., 147, 2083–2103.
We thank the reviewer for these references. We have modified the introduction text accordingly. Lines 40-57.

L120: suggest rephrase "during each pulse" – "during each of the 3 pulses". Or something like this. It took me a minute to understand why there are 3 numbers behind the hyphon.

We have rephrased. Lines 140-142.

L127: At which altitude to you spray?

All the particles are injected at the surface. We have stated this in the revised draft in line 149-150.

L137: Please explain why the TKE is used here as a metric for lateral dispersion. I would have thought that the length-scale of mesoscale organisation may be a better constraint?

The dispersion of the injected aerosol particles is driven by turbulence, and TKE is a standard measure of turbulence in the boundary layer. We have modified the text to mention this. Lines 160-162.

L145ff: I agree with the authors on this. The fact that you reach an area fraction of 1 suggests, that your results in the 5x scenarios are likely limited by domain size. This reduction of the analysis to ship track scales, should be made more clearly in the abstract. The current first sentence is ambiguous in this regard. This holds especially for all conclusions based on the Na=50mg-1 background experiments discussed in the following section.

We have removed the reference to ship tracks in the abstract. The limitations of the NA50 case is discussed in the revised manuscript. See earlier comments for details.

L157: „weakening of the updrafts (Fig. 3). Where do I see this in Fig. 3? What is included in your metric of TKE (i.e. subgrid and grid-scale components)

TKE represents the resolved component. The subgrid scale contribution is less than 1%. The reduction in TKE indicates the reduction in both horizontal and vertical components of the TKE. For clarity we have included the vertical velocity variance on the same plot.

L166ff: Why is the deepening during the first days a function of the difference in Na across the inversion in your simulations? This is not clear. Is it not rather that incloud Nd is not increased that much in the 1x experiments, such that entrainment feedbacks are not effective?

We believe the reviewer misunderstood our statement. The deepening is not affected by the difference in Na across the inversion, but the evolution of Na in the MBL is affected by the deepening. In the 1x cases, the difference in Na between the MBL and the FT is quite low. Thus, the effect of entrainment dilution on Na is quite weak. On the other hand, in the 5x cases, the difference in Na across the inversion height is very high and this reduces the Na in the MBL.

L175ff: Your explanations of the observed behaviour in LWP and fc in seeded experiments lack evidence. Please provide this (at least as part of this review).

We have included a time series plot showing SW absorption (Fig. 4). This figure along with the inversion height time series in Fig.2f explains the behavior of LWP and fc. Lines 197-199, 205-206.

L184: I agree with your statement, but please define the „onset of the transition“.

Done. Lines 134-136.

Fig4: Please add injection times like in Fig. 2. Hours 5, 13, 21 are unconventional. Why were they chosen? Also maybe you could add a timeline for shortwave absorption here to also provide partial proof for your arguments presented earlier (i.e. comment on L175ff).

In Fig. 4b, the vertical profiles are shown at intervals of 8 hours. On the first day, hour 13 is the time just before the start of the aerosol injection. We have included a figure showing the time series of SW absorption by the cloud layer.

Your time labels in Fig. 4b (i.e. starting at day 0) are different to the rest of the paper. Please adjust. Generally, each figure caption showing temporal slices with a periodic time axis should state the starting day for clarity. I also believe the day0 label in Fig.4b refers to the period marked under „day 1“ in Figure 2 and throughout the manuscript.

We have updated this figure. We have stated in Fig. 2 caption that all time series plots start from 10 am on day 1 to 10 am on day 4.

L189ff: I am not sure that your metric of BFI is really so usefull to diagnose decoupling in this case. For instance you simulate low BFI in periods following day 2 h12 (as low in the first 24 coupled hours) while the profile of Na+Nd clearly indicate decoupling. Other metrics have previously been

suggested in the literature that may show this more clearly?

As stated in the manuscript, a non-zero BFI is the minimum criteria required for decoupling. We agree that there are several other metrics for decoupling in the literature. However, none of them work well in the presence of precipitation. The vertical profiles of Na+Nd combined with BFI paints a clearer picture about the extent of decoupling within the MBL. The BFI is a good metric in comparing the strength of decoupling across cases at a certain instant in time. We have mentioned this in the Fig. 5 caption in the revised manuscript.

Fig5: The shift im time-axis from day-to-day (i.e. first 10, 20, 6h,16,02,...) makes it hard to look at the periodicity of the cloud-radiative effect. I would suggest a uniform periodic time axis.

We have updated the time axis in all the relevant plots.

L288: „changes in fc and are proportional to net injected aerosol concentration" The term „net injected" I find misleading. There is no sink in an injection. I would rephrase. Also, where do I see this?

We have replaced "net" with "total". After the injection of Cases 1x-110 and 1x-101 show similar cloud properties and similar cloud break up time. Both these cases have equal number of aerosols injected into the MBL before the onset of the transaction. This is mentioned in the discussion. Lines 381-390.

Fig. 8. Same questions and comments apply as in Fig. 4.

All questions addressed.

L348: rephrase „affected mainly by the total number of injected particles" to „ affected mainly by the total number of injected particles prior to the transition". I beleive your statement only holds under these conditions, as only then the number of injected particles is the same. Otherwise, this statement cannot be true, as temporarily the number of injected aerosol will by design be different between 1x-101 and 1x110.

We agree with the reviewer. We have rephrased the text. Lines 379-380.

L367: „earlier transition" what transition? It already starts out in the cumulus state in 0x-000 right? I think the suggested definition of the transition in an earlier comment can help clarify this point.

We have rephrased the text and have included a definition of the transition. See earlier comments.

L391: According to table 1 there is no saturation in the Twomey effect though. There is a $\sim 20$ W/m2 difference between 5x-200 and 5x-220 on day 3.

We have rephrased the text. Lines 425-429.

**Reviewer 2:**

Summary:

The authors investigated the impact of injected aerosols on the transition from stratocumulus to cumulus clouds in the north-east Pacific. They revealed that aerosol injection delays the stratocumulus-to-cumulus transition, with the extent of the delay directly proportional to the number of aerosol particles injected into the marine boundary layer, ultimately influencing cloud radiative effects in both pristine and polluted systems. Overall, I suggest publishing it after addressing the comments below.

We thank the reviewer for the positive evaluation of our paper and for the constructive comments. In our response below, the reviewer comments are in black and our responses are in blue.

**Major comment:**

Injections time are chosen to be the same stage during diurnal cycle. Why is that? How will it impact the results if injection at different time during the diurnal cycle?
We carried out another simulation for NA150. We repeated the case 5x-100 but the perturbation was delayed by 5 hours (around sunset time on the first day). We see that there is no major difference between the cloud properties in regular 5x-100 and the delayed version (see Fig. 2). A small difference is evident towards the end of the simulation between these cases, and could be related to the variance associated with the onset of strong precipitation on the evening of day 3. This supports our conclusion that the timing of the aerosol pulse, to the leading order, does not substantially affect the cloud brightening in NA150. We have not explored the sensitivity of timing in NA50 as earlier works have shown that a strongly precipitating Stratocumulus cloud layer is sensitive to timing of the aerosol pulse (Prabhakaran et al 2023, JAS).

**Minor comments:**

Line 75: why? the deficiency with small LES domain is the lack of feedback from larger scale, but the Lagrangian LES does not solve this problem, even the large-scale variabilities are presented by forcings. On top of that, whether forcing is good enough is a new problem.
We thank the reviewer for this comment. We agree with the reviewer and have modified the introduction to reflect the concerns raised by the reviewer. Lines 90-93.

Line 82: recommend 1-2 sentence to clarify what does "bin-emulating, bulk microphysical model" mean.
This means that bin-by-bin mass transfer rates are used for modeling the collision-coalescence process. Line 102-104.

Line 90: "The two modes are separated by a threshold value of 25 .m in radius." reference? It is larger than the conventional 12 or 13 micro as the start of auto conversion. Why?
We have included two references - Kessler 1969 and Khairoutdinov and Kogan 2000). Note that 12-13 um threshold is for effective radius of the droplet size distribution. Typically, a size of over 20 um in radius is required for efficient collision.

Line 96: "In the applied modeling framework, cloud processing of aerosol affects the number concentration of aerosol but not the shape of the distribution." Reference?
We have included a reference - Feingold et al 1998.

Line 102-104: "the results are highly relevant in terms of the injection-related modification to and the subsequent adjustments of LWP and fc, which together determine the degree of Nd cloud brightening." Besides LWP and fc, how about cloud base height, cloud top height and cloud depth?
The focus of this study is cloud albedo enhancement. The cloud base and top heights do not directly contribute to cloud albedo, the changes in cloud depth are relevant and represented through the LWP.

Line 130: "the plume", what does the "plume" mean? Injected aerosol plume? Or cloud plume? I don't think I understand why the authors can use aerosol concentration to represent a plume, that usually use cloud optical thickness or cloud albedo to identify. They are two quite distinguished concept. Also, does it make more sense to use na+nd to identify plume than solely na?
We are referring to the injected aerosol plume. We have clarified this in the revised text (Lines 153-157). In the earlier version of the manuscript, we have clarified why we use Na instead of optical thickness. In the current version, Lines 153-158 state why we do not use optical thickness.

Line 135-137: which figure are you referring to representing spread rate?
Figure 2a. We have stated this in the revised version.

Line 151-152: why? Why not separate the plume and background region?
We did not separate the plume and the background as it did not add much value to the key results in the article. For MCB applications, the entire scene is relevant.

Line 156-158: "On day 3, decreases by about 40% by midday due to (i) collision-coalescence and precipitation losses, Nd and (ii) reduced aerosol activation rate due to the weakening of the updrafts (Fig. 3) due to precipitation evaporation and SW absorption." Figure 3 does not show updraft. Also, how do you know the activation fraction is reduced? If that is inferred, soften the sentence to reflect that.
We have included the vertical velocity variance time trace to Fig. 3. We have softened the sentence regarding activation as the reduction in activated fraction is inferred. Lines 179-180.

Line 164: remove the bracket
Done.

Many places explain the phenomena by entrainment but there is no direct reflection of the cloud top entrainment strength. How about adding it to the figures?
Several earlier studies have established that in the absence of precipitation the changes in LWP in response to changes to $N_d$ is from enhanced entrainment. A measure of entrainment velocity is the changes to the inversion height as a function of time. The inversion height time trace is shown for all the simulations. In the revised draft we have mentioned the relation between entrainment rate and the inversion height time series. Lines 197-199.